# Emerging Therapeutic Strategies for Diffuse Intrinsic Pontine Glioma: A Systematic Review

**DOI:** 10.3390/healthcare11040559

**Published:** 2023-02-13

**Authors:** Shahrukh Farrukh, Shagufta Habib, Amna Rafaqat, Zouina Sarfraz, Azza Sarfraz, Muzna Sarfraz, Karla Robles-Velasco, Miguel Felix, Ivan Cherrez-Ojeda

**Affiliations:** 1Department of Research, Khawaja Muhammad Safdar Medical College, Sialkot 51310, Pakistan; 2Department of Research, University Medical and Dental College Faisalabad, Faisalabad 38800, Pakistan; 3Department of Research and Publications, Fatima Jinnah Medical University, Lahore 54000, Pakistan; 4Department of Pediatrics and Child Health, The Aga Khan University, Karachi 74000, Pakistan; 5Independent Researcher, Lahore 54000, Pakistan; 6Department of Allergy, Immunology and Pulmonary Medicine, Universidad Espíritu Santo, Samborondón 092301, Ecuador; 7Department of Internal Medicine, New York City Health + Hospitals, Lincoln, The Bronx, NY 10451, USA

**Keywords:** diffuse intrinsic pontine glioma, CNS, tumor, therapies, palliative, quality of life, advances

## Abstract

Background: Of all central nervous systems tumors, 10–20% are located in the brainstem; diffuse intrinsic pontine glioma (DIPG) is diagnosed in 80% of them. With over five decades of clinical trial testing, there are no established therapeutic options for DIPG. This research article aims to collate recent clinical trial data and provide a landscape for the most promising therapies that have emerged in the past five years. Methods: PubMed/MEDLINE, Web of Science, Scopus, and Cochrane were systematically searched using the following keywords: Diffuse intrinsic pontine glioma, Pontine, Glioma, Treatment, Therapy, Therapeutics, curative, and/or Management. Both adult and pediatric patients with newly diagnosed or progressive DIPG were considered in the clinical trial setting. The risk of bias was assessed using the ROBINS-I tool. Results: A total of 22 trials were included reporting the efficacy and safety outcomes among patients. First, five trials reported outcomes of blood–brain barrier bypass via single or repeated-dose intra-arterial therapy or convection-enhanced delivery. Second, external beam radiation regimens were assessed for safety and efficacy in three trials. Third, four trials administered intravenous treatment without using chemotherapeutic regimens. Fourth, eight trials reported the combinations of one or more chemotherapeutic agents. Fifth, immunotherapy was reported in two trials in an adjuvant monotherapy in the post-radiotherapy setting. Conclusion: This research article captures a clinical picture of the last five years of the direction toward which DIPG research is heading. The article finds that re-irradiation may prolong survival in patients with progressive DIPG; it also instills that insofar palliative radiotherapy has been a key prognostic choice.

## 1. Introduction

Central nervous system (CNS) tumors have a higher mortality rate among all cancers in US children aged 1–19 years [1,2,3]. Of all CNS tumors, 10–20% are located in the brainstem, with diffuse intrinsic pontine glioma (DIPG) diagnosed in 80% of them [4]. The prognosis is dismal with DIPG with the overall survival rate being lower than 10% at 2 years [5]. The median survival rates are less than 12 months and the 5-year survival rates are below 2% [6,7]. Radiation therapy is the current standard of treatment, yet it remains a palliative option as radiotherapy only temporarily relieves symptoms [8]. DIPG has been classified histopathologically as high-grade astrocytoma (HGA) with most cases being consistent with histological grade III or IV (anaplastic astrocytoma or glioblastoma, respectively) but certain less aggressive cases are histologically classified as grade II (diffuse astrocytoma) [9]. More recently, the World Health Organization (WHO) classified DIPG as diffuse midline gliomas with histone H3K27M mutation [9]. With over five decades of clinical trials exploring different chemotherapy and radiotherapy regimens, there are no promising therapeutic options in DIPG [10]. Several clinical trials of systematic therapies have been tested or are ongoing [11].

Certain therapies have recently gained traction for their potential efficacy in DIPG [12]. Thereby, this research article aims to collate data from recent clinical trials and provide a landscape for the most promising therapeutic options that have emerged in the last five years.

## 2. Materials and Methods

### 2.1. Search Strategy

A comprehensive systematic search was conducted using the following databases, adhering to Preferred Reporting Items for Systematic Reviews and Meta-Analyses (PRISMA) Statement 2020 guidelines: PubMed/MEDLINE, Web of Science, Scopus, and Cochrane. The search was conducted from 1 January 2017, until 16 October 2022, without any language limitations (non-English studies were translated to English using Google Translate). Applying the Boolean (and/or) logic, the following keywords were applied: Diffuse intrinsic pontine glioma, Pontine, Glioma, Treatment, Therapy, Therapeutics, curative, and/or Management. The titles and abstracts of shortlisted studies from the given databases were screened independently by two mid-career authors (Z.S., A.S.). In the screening phase, the reference lists were also reviewed, in line with the umbrella methodology to ensure no data were omitted. In the case of any disagreements, a third author (I.C.-O.) was present to resolve them and to reach a consensus. Cohen’s coefficient of the inter-reviewer agreement was computed using Statistical Package for Social Sciences (SPSS, v.25, IBM).

### 2.2. Inclusion and Exclusion Criteria

The inclusion criteria comprised clinical trials enrolling both adult and pediatric patients of any gender with newly diagnosed or progressive DIPG. The treatment could consist of any of the following: (i) blood–brain barrier bypass (intra-arterial therapy or convection enhanced delivery), (ii) external beam radiation, (iii) intravenous treatment regimens without administering chemotherapeutic agents, (iv) combination regimens with the use of one or more chemotherapeutic agents.

Studies intervening with only surgical procedures were not included. Further, cohorts (retrospective/prospective), case series, case reports, systematic reviews and meta-analyses, brief reports, and letters to editors were excluded.

### 2.3. Data Extraction (Selection and Coding)

Two mid-career authors (Z.S., A.S.) independently extracted the data from the included trials into a spreadsheet. The third author (I.C.-O.) was present for any disagreements. The pair identified the trials and treatments of the screened studies. Once the independent review of the studies was conducted, the third author (I.C.-O.) assessed the extracted domains from the spreadsheet and conducted the final review against the inclusion criteria.

The data were extracted as follows: Author, Year, Title, Journal, Country, Study design, Inclusion criteria, Intervention given, Method of administration; Number of patients with DIPG, Age at diagnosis (in years), Sex (percentage of males), Previous treatment, Outcome measures; Median OS (in months), Median EFS/PFS (in months), Radiological response (percentage), Clinical improvement (proportion), Tolerance and safety, and Steroid use discontinuation.

Individual study data were prepared in a presentable format during the inclusion phase and the concluding remarks were also added. EndNote X9 (Clarivate, London, UK) was the software used to omit duplicates during the study selection process. In addition, Mendeley (Elsevier, Amsterdam, The Netherlands) was used for bibliographic management.

### 2.4. Risk of Bias (Quality Assessment)

The Risk of Bias in Non-randomized Studies of Interventions (ROBINS-I) tool was used to assess the risk of bias in the included trials. This tool comprised seven domains. (1) Bias due to confounding; (2) bias due to selection of participants; (3) bias in classification of interventions; (4) bias due to deviations from intended interventions; (5) bias due to missing data; (6) bias in the measurement of outcomes; (7) bias in the selection of the reported result.

Domain-level judgments about the risk of bias were classified as the following: (1) low risk; (2) moderate risk; and (3) serious risk. The traffic light plot of bias assessment and the weighted summary plot of the overall type of bias encountered is illustrated in Section 3.6.

## 3. Results

During the phase I, the identification phase, a total of 1249 studies were identified. Of these, 234 duplicates were removed. In phase II, the screening phase, 1015 study titles and abstracts were screened, of which 897 were omitted as they did not warrant inclusion against the inclusion criteria. Subsequently, 118 full-text studies were reviewed and assessed for eligibility. Of these, 96 studies were excluded as 28 were non-human studies, 43 of them were model/experimental papers, and 25 were not clinical trials. In phase III, the inclusion phase, a total of 22 trials were included (Figure 1). Kappa’s score was calculated to be 0.91.

A total of 22 trials were found that reported efficacy and safety outcomes among patients with newly diagnosed or progressive DIPG (Table 1, Table 2 and Table 3). First, trials that reported bypassing of the blood–brain barrier (BBB) are reported. In five phase I trials, authors evaluated outcomes of blood–brain barrier (BBB) bypass via single or repeated-dose intra-arterial therapy [13] or convection-enhanced delivery (CED) [14,15,16,17].

Second, trials that explored different external beam radiation regimens are elaborated. In two randomized controlled trials [18,19], the authors explored the efficacy and safety of different radiotherapy regimens including conventionally fractionated and hypofractionated radiotherapy in newly diagnosed DIPG. Re-irradiation at three dose levels among patients who had received initial radiotherapy ≥10 months ago was compared with patients with progressive DIPG in a phase I/II trial [20].

Third, trials are listed that administered intravenous treatment regimens without administering chemotherapeutic agents. In a phase III trial, authors evaluated the outcomes of epidermal growth factor receptor (EGFR) inhibitors, nimoztuzumab with radiotherapy in newly diagnosed DIPG patients [21]. In a phase I/II trial, the authors evaluated the outcomes of vorinostat given concomitantly and as adjuvant to radiotherapy in newly diagnosed DIPG [22]. Adavosertib, a Wee 1 kinase inhibitor, was given with cranial radiation therapy (CRT) in newly diagnosed DIPG in a phase I trial [23]. A phase II trial administered EBT and valproic acid (VPA), an anti-convulsant, followed by bevacizumab, an anti-vascular endothelial growth factor, and VPA in newly diagnosed DIPG patients [24].

Fourth, trials that evaluated combination regimens with the use of one or more chemotherapeutic agents are listed. One phase II trial with newly diagnosed DIPG combined radiotherapy and cefuximab, an epidermal growth factor receptor (EGFR) inhibitor, followed by cefuximab and irinotecan, a topoisomerase I inhibitor [25]. Another phase I/II trial combined erlotinib, an EGFR inhibitor, with bevacizumab, a vascular endothelial growth factor (VEGF) inhibitor, and irinocetan among patients with progressive DIPG [26]. Two trials explored outcomes of ribociclib (kinase inhibitor) concomitantly with radiotherapy and everolimus (kinase inhibitor) (phase I trial [27]) and as adjuvant monotherapy (in phase II [28]) in newly diagnosed patients with DIPG. One phase I/II trial administered concomitant EBT and veliparib, a PARP inhibitor, followed by veliparib and temozolomide, an alkylating agent [29]. Capecitabine, an alkylating agent, was given in combination with EBT and as adjuvant monotherapy in newly diagnosed DIPG patients in a phase II trial [30]. Gemcitabine, a nucleoside metabolic inhibitor, was combined with initial EBT among newly diagnosed DIPG patients in a phase I/II trial [31]. One trial explored the maximum tolerated dose (MTD) and efficacy of cabazitaxel, a microtubule inhibitor, as adjuvant monotherapy in progressive DIPG patients in a phase I/II dose-escalating trial [32].

Fifth, trials that administered immunotherapeutic agents as adjuvant monotherapy after radiotherapy are reported. One phase II trial administered pomalidomide, an immunomodulatory drug, among patients with progressive DIPG. One phase I trial administered pelareorep, an immunomodulatory oncolytic virus, combined with sargramostin, a recombinant human granulocyte-macrophage colony-stimulating factor, in patients with progressive DIPG.

### 3.1. Bypassing the Blood–Brain Barrier

#### 3.1.1. Intra-Arterial Therapy

McCrea et al. explored the tolerability and efficacy of super selective intraarterial cerebral infusion (SIACI) among 10 patients with DIPG who had all previously received radiotherapy, as well as other systemic therapies [13]. Mannitol (12.5 mL of 20%) was administered to disrupt the blood–brain barrier (BBB), followed by bevacizumab (15 mg/kg), a vascular endothelial growth factor A (VEGF-A) inhibitor, and cetuximab (200 mg/m^2^), an epidermal growth factor receptor (EGFR) inhibitor, respectively [13]. The treatment and technique were well-tolerated with no dose-limiting toxicities. In terms of efficacy, the median OS was 17.3 months, which is higher than that for historical controls despite DIPG patients being heavily treated [13]. This method of delivery warrants further investigation to establish efficacy in DIPG patients [13].

#### 3.1.2. Convection-Enhanced Delivery

Heiss et al. evaluated the outcomes including the safety and tolerability of single-dose IL13-PE38QQR, a recombinant cytotoxic chimera of human interleukin 13 (IL-13), and the enzymatic portion of pseudomonas exotoxin A, infused via single-catheter CED (0.125 μg/mL) into 5 patients with progressive DIPG [14]. The intervention was safe and well-tolerated which occurred due to infusion-related brainstem edema [14]. There was a temporary improvement in clinical and radiological status in 2 patients (40%) and their dose was escalated to 0.25 μg/mL. CED-supported delivery of IL13-PE did not reach optimal volumes in the tumor and only temporary anti-tumoral effects were observed in 2 patients [14].

Pérez-Larraya et al. determined the safety and efficacy outcomes of single-dose CED infusion of DNX-2401, an oncolytic adenovirus that only replicates in tumor cells, through the cerebellar peduncle, followed by radiotherapy in 12 newly diagnosed DIPG patients [15]. The median OS was favorable at 17.8 months; 11 of the 12 patients had a reduction or stabilization of tumor size but certain adverse events (hemiparesis in 1 patient and tetraparesis in 1 patient) were reported [15].

Bander et al. evaluated the efficacy and safety of ≥2 doses of CED, via the supratentorial trajectory with intraprocedural stereotactic placement using MRI guidance, infusing I-8H9 monoclonal antibody among 7 DIPG patients who previously received radiotherapy [16]. Sequential CED infusions were well-tolerated and the second infusion significantly reduced radial error and absolute tip error compared to the first infusion [16]. This was a phase I trial whereby Bander et al. lay support for repeated CED in DIPG patients for further evaluation of improved survival rates [16].

Majzner et al. similarly delivered two doses of disialoganglioside GD2-directed chimeric antigen receptor (CAR) T cell; the first dose administered intravenously and the second dose delivered intracerebroventricularly to 3 patients with K27M mutation in genes encoding histone H3 (H3K27M) at any stage. All patients had received radiotherapy ≥ 6 months before enrollment. Of the three patients, two had improvement radiologically and one had tumor progression [17]. All patients had infusion-related toxicity which was reversible with intensive support [17]. With the necessary management algorithm for tumor inflammation-associated neurotoxicity (TIAN), delivery of CAR T-cell therapy via CED is a promising treatment option that may be explored further in trial settings [17].

### 3.2. Different Radiotherapy Regimens

#### 3.2.1. Hypofractionated Radiation Therapy

Zagloul et al. confirmed the noninferiority of hypofractionated (HF) radiation therapy with three arms, arm 1 received HF therapy of 39 Gy in 13 fractions, arm 2 received HF therapy of 45 Gy in 15 fractions, and arm 3 received conventional fractionation (CF) of 54 Gy in 30 fractions [18]. The median OS was the highest in low-dose HF across arm 1 (9.6 months) followed by CF in arm 3 (8.7 months) and high-dose HF in arm 2 (8.2 months). Younger age (2–5 years) had a higher prognosis with lower HF dose given in arm 1 which was not found in arm 2 recipients [18].

Izzuddeen et al. compared the efficacy and tolerability of CF radiotherapy and low-dose HF radiotherapy (39 Gy in 13 fractions) with concurrent and adjuvant temozolomide, an alkylating agent [19]. The group that received HF radiotherapy and temozolomide did not show any improved survival rates (12 months vs. 11 months) and there was an increase in grade 3/4 hematological toxicity (*n* = 5, 28%) [19].

#### 3.2.2. Re-Irradiation

Amsbaugh et al. found clinical improvement and improved quality of life with conventionally fractionated re-irradiation among patients who had ≥10 months from the end of initial radiotherapy [20]. The lowest dose arm of 24 Gy in 12 fractions had the highest utility and is considered safe and effective for re-irradiation in DIPG patients [20].

### 3.3. Non-Chemotherapeutic Agent Regimens

Fleischhack et al. assessed the safety and efficacy of intravenous nimotuzumab, an anti-EGFR humanized monoclonal antibody, combined with external beam radiotherapy (EBT) among treatment-naïve patients with DIPG diagnosed in the last 3 months. The intervention was well-tolerated and no adverse events were observed, such as those produced by other EGFR-targeting agents e.g., severe acneiform rash, hypokalemia, or hypomagnesemia [21]. Nimotuzumab administered concomitantly and continued after EBT had comparable efficacy to intensive chemotherapy and EBT among newly diagnosed DIPG e.g., median OS of 9.4 months while offering the benefit of being administered in outpatient settings and associated with reduced hospitalization stays [21].

Su et al. reported the efficacy and tolerability of Vorinostat, an oral histone deacetylase inhibitor, given with initial radiotherapy and as monotherapy afterward [22]. While it was well tolerated, there were no survival benefits in patients with newly diagnosed DIPG [22].

Mueller et al. determined the tolerability of Adavosertib, an orally administered blood–brain barrier penetrant, Wee 1 kinase inhibitor when combined with cranial radiation therapy (CRT) among patients with newly diagnosed DIPG [23]. While the treatment was well-tolerated, there was no survival benefit with a median OS of 11.1 months [23].

Su et al. explored the tolerability and efficacy of valproic acid (VPA), an anti-convulsant, and radiation, followed by VPA and bevacizumab, an anti-VEGF humanized monoclonal antibody, among patients with newly diagnosed DIPG in a phase II trial [24]. While the concomitant use of VPA and bevacizumab was well-tolerated, there was no improvement in survival outcomes with a median OS of 10.3 months [24].

### 3.4. Chemotherapeutic Agent Regimens

Macy et al. evaluated the efficacy and safety of cetuximab, anti-EGFR humanized with concurrent radiation therapy followed by cetuximab and irinotecan, a topoisomerase I inhibitor, among patients with newly diagnosed DIPG [25]. While there was some improvement in PFS, the median OS was 12.1 months and the therapy regimen does not warrant further investigation [25].

El-Khouly et al. conducted a phase I/II trial to determine the safety, tolerability, and efficacy of bevacizumab, a VEGF inhibitor, irinotecan, a topoisomerase I inhibitor, and Erlotinib, an EGFR inhibitor, among 9 patients with progressive DIPG. The median OS was 13.8 months which was higher than that of historical controls for this subset of patients and the regimen was well-tolerated [26].

DeWire et al. conducted a phase I trial to determine the tolerability and efficacy of Ribociclib (CDK4/6-inhibitor) and Everolimus (kinase inhibitor) for patients newly diagnosed with DIPG within 30 days of receiving a 10% standard dose of radiotherapy [27]. The treatment was well-tolerated and apparently improved median OS to 13.9 months; however, when two patients who were less than 3 years at diagnosis e.g., associated with better prognosis were removed, the median OS decreased to 10.8 months which suggests no additional efficacy of treatment [27].

DeWire et al. also conducted a phase I/II trial to identify the safety, feasibility, and early efficacy of ribociclib (CDK4/6-inhibitor) in 9 newly diagnosed DIPG patients [28]. Ribociclib adjuvant monotherapy post-radiotherapy had improved median OS (16.1 months) [28]. Yet, its safety is not clear as there was increased tumor necrosis volume in 4 patients (40%) warranting further volumetric analyses of necrosed tumors [28].

Baxter et al. conducted a phase I/II trial of 65 patients with newly diagnosed DIPG who received concomitant veliparib (PARP inhibitor) and radiation therapy followed by veliparib and temozolomide (alkylating agent) [29]. While the treatment was generally well-tolerated with limited DLTs, there were no survival benefits categorized as 1-year and 2-year survival rates of 37.2% and 5.3%, respectively [29].

Kilburn et al. conducted a phase II trial with capecitabine (alkylating agent) given concomitantly with radiotherapy followed by adjuvant capecitabine among 44 patients with newly diagnosed DIPG [30]. There was no survival benefit (OS and PFS were comparable to historical controls) with the treatment regimen but it was well-tolerated [30].

Zanten et al. conducted a phase I/II trial to determine the efficacy, safety, and tolerability of gemcitabine, a nucleoside metabolic inhibitor, during radiotherapy among 9 children with newly diagnosed DIPG [31]. The treatment was well-tolerated and a dose of up to 200 mg/m^2^/once weekly with radiotherapy was safe [31]. There were, however, no survival benefits with a median OS of 12.4 months and 8.7 months in intermediate- and high-risk patients [31].

Manley et al. explored the maximum tolerated dose (MTD), safety, and efficacy of Cabazitaxel, a chemotherapeutic agent, in patients with progressive treatment-refractory DIPG in a phase I/II dose-escalating trial [32]. The maximum tolerated dose (MTD) was found to be 30 mg/m^2^ and was well-tolerated. There was no anti-tumor activity with the MTD and there was no improvement in survival outcomes [32].

### 3.5. Immunotherapy

Fangusaro et al. conducted a phase II trial and determined the efficacy and safety of Pomalidomide, an immunomodulatory drug, among 9 patients with progressive DIPG [33]. There were no favorable survival outcomes of pomalidomide monotherapy with no objective response (OR) or long-term stable disease (LTSD) found in the patients and a median OS of 3.78 months [33].

Schuelke et al. determined the safety of sargramostin, a recombinant human granulocyte-macrophage colony-stimulating factor, and pelareorep, an immunomodulatory oncolytic virus, among 2 patients with progressive DIPG in a phase I trial [34]. While the treatment was well-tolerated, the sample size was too small to evaluate survival rates; the treatment warrants further investigation for efficacy [34].

### 3.6. Risk of Bias Synthesis

Overall, 19 studies (86.4%) had a low risk of bias, 2 (9.1%) had moderate risks and 1 (4.5%) had a serious risk of bias (Figure 2). On noting bias due to confounding, 17 studies (77.3%) had a low risk of bias, 4 (18.2%) had moderate risk, whereas 1 (4.5%) had a serious risk of bias. When assessing bias due to the selection of participants, a total of 14 studies (63.6%) had a low risk of bias, whereas 7 (31.8%) had a moderate risk of bias and 1 (4.5%) had a serious risk of bias. Noting the bias in the classification of interventions, 20 studies (90.9%) had low risks of bias while 2 studies (9.1%) had a moderate risk of bias. Bias due to deviations from intended interventions had low risk in 21 studies (95.5%) and moderate risk in 1 study (4.5%). On noting bias due to missing outcome data, 18 studies (81.8%) had a low risk of bias, and 2 studies each (9.1%) had moderate and serious risks of bias. Assessment of bias in the measurement of outcomes yielded 18 studies (81.8%) with a low risk of bias and 4 studies (18.2%) with a moderate risk of bias. The risk of bias in the selection of the reported result was low in 12 studies (54.4%), moderate in 9 studies (40.9%), and serious in 1 study (4.5%) (Figure 2).

## 4. Discussion

This research article aimed to collate evidence from all trials conducted in the last five years to evaluate the efficacy and safety of different treatments for DIPG. We assessed 22 trials and five key therapeutic regimen themes emerged including blood–brain barrier (BBB) bypass (intra-arterial delivery or convection-enhanced delivery (CED)), radiotherapy regimens, non-chemotherapeutic agent regimens, chemotherapeutic agent regimens, and immunotherapy. The 22 trials included in this study are likely to constitute all available clinical trial evidence, due to the robust search strategy and rigorous screening process.

Of the 5 trials reporting BBB bypass techniques, 3 reported overall survival (OS), 4 reported radiological responses, 3 reported clinical improvement, all 5 reported tolerance and safety, and 3 reported steroid discontinuation. All 3 trials that explored radiotherapy regimens and doses did report OS, 2 trials reported progression-free survival (PFS), 1 reported radiological response, 1 reported clinical improvement, and all 3 reported tolerance and safety outcomes. Four trials that explored non-chemotherapeutic agents and OS were reported across all 4 trials, PFS in 1 trial, event-free survival (EFS) in 2 trials, radiological response in 3 trials, and safety/tolerance in all 4 trials. A total of 8 trials administered chemotherapeutic agents in combination with other therapeutics and all 8 trials reported OS, 5 trials reported PFS, all 8 trials reported radiological response, 3 trials reported clinical improvement, and all 8 reported tolerability/safety. Lastly, of the 2 trials that administered immunotherapy agents, both reported OS, 1 reported PFS, 2 reported radiological response, 1 reported clinical improvement, and both identified safety/tolerance. There were differences among studies, even within the same theme, with regard to outcome measures, specifically radiological response.

All therapeutic agents were initiated at different time points in the natural clinical course of DIPG. The most commonly observed subset of patients was newly diagnosed across 12 trials, followed by progressive DIPG in 8 trials. One trial enrolled patients at any stage and 1 trial did not specify the stage of DIPG. Patient demographics were similar across the 22 trials. Age at diagnosis was primarily mid-childhood. The gender ratio was somewhat well-balanced, ranging from 35–65% across the studies.

### 4.1. Intra-Arterial Delivery

Superselective intraarterial cerebral infusion (SIACI) improved survival rates (median OS: 17.3 months) when offered to patients with progressive DIPG. SIACI offers an advantage over intravenous drug delivery through selective blood–brain barrier (BBB) opening. Our synthesis supports the intraarterial delivery of cetuximab and bevacizumab with the initial administration of mannitol to increase the absorption of the drugs. As we found support for safe and well-tolerated repeated CED infusions, further trials can consider expanding the number of patients and determining the efficacy of SIACI. A phase I trial (NCT05271240) is underway that is planning enrollment of 432 patients with glioblastoma multiforme (GBM) and comparing repeated mannitol-infusion followed by SIACI of bevacizumab with temozolomide and standard radiation to temozolomide and standard radiation only. As data are still emerging regarding the safety and tolerability of intra-arterial delivery, further trials can consider using a labeling agent to assess drug delivery distribution. Another consideration is to identify molecular targets with a biopsy to optimize the agent of choice. With evidence of safe bypassing of the BBB, it is of note to consider targeting tumor cells based on the biology e.g., EGFR and/or VEGF positive.

### 4.2. Convection-Enhanced Delivery

CED is an emerging therapy for DIPG due to its ability to bypass the BBB and deliver pertinent doses of treatment in relevant brain volumes. The agents tested in clinical trials via CED were ^124^I-8H9 [16], IL13-Pseudomonas toxin [14], DNX-2401 (an oncolytic virus) [15], and GD2-CAR T cells [17]. Of the 4 trials evaluating the role of convection-enhanced delivery (CED) in the treatment of DIPG, there were somewhat favorable outcomes based on different techniques, agents used, and stages of DIPG. Intra-tumoral infusion with an oncolytic virus had the most favorable outcomes with a median OS of 17.8 months in newly diagnosed DIPG but there was an increased risk of adverse events related to infusion-related brainstem edema. Augmentation of CED-infused pharmacological agents gained support from observational studies such as Tsvankin et al. who found improvement in median OS with CED of dasatinib, a tyrosine kinase inhibitor, in a transgenic H3.3K27M mutant murine model [35]. It remains unclear which pharmacological agent offers the highest survival rates and it is reasonable to consider CED to have a plateau effect given its ability to target localized disease.

Repeated CED infusions were well-tolerated but neurological signs and symptoms are present. Hollingworth et al. [36] measured infusion-related side effects of CED in DIPG with the Pontine Neurological Observation Score (PONScore), a standardized tool with a 57-point scale, to determine their frequency and recovery during infusion. As CED is gaining preliminary support from the literature for its efficacy in DIPG, there is a gap in standardized documentation of its side effects for which a scale such as PONScore [36] can highlight the nature and timing of neurological injury during infusion. Further clinical trials must consider it imperative to consider meticulously documenting infusion-related side effects to address patient outcomes. As of now, CED is being tested in early phase clinical trials for treatment with DIPG which has the potential to control the regional disease. However, CED as a standalone may not be able to control spread to distant areas e.g., outside the brainstem [37] or leptomeningeal involvement [38], by nature of its delivery and it may be advantageous to consider combination approaches e.g., craniospinal radiation or intrathecal delivery [39], to meaningfully improve survival rates in DIPG patients who already have a dismal prognosis.

### 4.3. Radiotherapy

The mainstay of treatment for DIPG is conventionally fractionated radiation therapy (RT), delivered across a 6-week period. However, such RT only transiently improves symptoms without prominent survival benefits. Hypofractionated RT regimens did not demonstrate survival benefits across two trials in our study but may provide temporary relief. In a trial, re-irradiation did improve survival outcomes and quality of life among DIPG patients who had received initial radiotherapy ≥ 10 months ago (strongest support for low-dose conventional RT at 24 Gy in 12 fractions). Gallitto et al. [8] corroborate the findings of the trial and suggests re-irradiation at first progression as an effective palliative therapy with a mean increase of ~3 months to OS compared to controls. Combination immunotherapy agents (PD-1 inhibitor, nivolumab) [40] with re-irradiation have also shown improved life spans in progressive DIPG patients. Overall, certain patients can tolerate re-irradiation, have reduced symptoms, and improve survival rates by a few months. Current trials have compared different radiation doses and fractionation for re-irradiation; support for conventional fractionation and low-dose radiation is found in clinical trials. Further clinical trials can address the frequency of re-irradiation, the gap between radiotherapy, and optimal radiation dose and fractionation for children with progressive DIPG [41].

### 4.4. Other Regimens

Other trials explored the efficacy of radiotherapy with neoadjuvant non-chemotherapeutic interventions (nimotuzumab, bevacizumab, adavosertib, and vorinostat) with or without adjuvant therapy afterward in newly diagnosed DIPG patients and found no additional survival benefit compared to historical controls. Similarly, there was no significant improvement in survival outcomes with a neoadjuvant chemotherapeutic alkylating agent (temozolomide) or anti-metabolites (capecitabine, gemcitabine, cabazitaxel) in combination with radiotherapy, and other agents (cetuximab, an EGFR inhibitor, and veliparib, a PARP inhibitor). Moreover, immunotherapy regimens did not improve median OS with pomalidomide and pelareorep combined with sargramostim. There were, however, two trials by DeWire et al. [27,28] that found some improvement in median OS with combination regimens in newly diagnosed DIPG patients: median OS of 13.9 months with ribociclib and everolimus, both kinase inhibitors, within 30 days of receiving 10% standard radiation doses, and median OS of 16.1 months with ribociclib together with radiotherapy and adjuvant monotherapy. There is emerging support for ribociclib, a CDK4/6 inhibitor, together with targeted radiotherapy among treatment-naïve DIPG patients. However, in both these trials, there was a prominent frequency of ≥grade 3 events which may be due to the treatment. Another trial [26] administered bevaziumab, a VEGF inhibitor, erlotinib, an EGFR inhibitor, and irinotecan, a topoisomerase I inhibitor among progressive DIPG patients. Of note, this trial indicated that anti-EGFR agents and anti-VEGF antibodies when combined with chemotherapy have some additive antitumor activity (median OS of 13.8 months) and are well-tolerated. These findings provide support for the collaboration of anti-VEGF and anti-EGFR for inhibiting tumor growth and angiogenesis in aggressive DIPG, combined with chemotherapy.

### 4.5. Potential Molecular Targets

The lack of targeted therapies for DIPG is in part due to the lack of routine biopsies conducted in these tumors. Neurosurgeons have been reluctant to perform biopsies due to more risks than direct benefits to the patients. As molecular genetic techniques are expanding in oncology, many centers have begun conducting stereotactic biopsy to support ongoing research which requires molecular characterization and potentially druggable targets toward more individualized treatments [42,43]. About 80% of all DIPG cases have a specific point mutation that results in the substitution of lysine 27 on the amino-terminal tail with methionine (H3K27M) in histone isoforms H3.1 or H3.3, encoded by genes *HIST1H3B* and *H3F3A* respectively [44,45,46]. There are subtle differences in prognosis and outcomes with both; H3.1 histone mutations have a better prognosis and this has been recognized by the World Health Organization of CNS tumors [9,47]. Along with these histone modifications, molecular profiling has enumerated numerous targets for therapeutic interventions. These include ACVR1 mutations (~30% of DIPG tumors) and co-occur with H3.1 and TP53 mutations (~22–40% of DIPG tumors) which co-occur with PDFR amplification [48,49,50]. PDFR amplification (PDFRA) is common and present in nearly 1/3rd of high-grade gliomas; when co-segregated with H3.3 mutations (H3.3K27M), these tumors are clinically aggressive [51]. PDFRA combined with PIK3R1 and PIK3CA are drivers of the PI3K pathway which contributes to aggressive DIPG [48,52].

### 4.6. Strengths and Future Directions

The 22 studies in this study constitute the latest clinical trends for DIPG therapeutics with an inclusive search strategy and rigorous screening process. Two independent reviewers screened the full text and there was 91.6% agreement. As part of the search strategy, an umbrella review methodology was also applied which revealed four potential papers, though none were included. Another strength was the double-checking of data entry by the second reviewer. The conclusions made from this study are based on all the latest available evidence from clinical trials in the past 5 years. Therefore, the data included in this study are not observational, which means that the quality of included studies is not compromised. Lastly, the study was flexible in terms of eligibility criteria in relation to the nature of treatment which allowed for a comprehensive synthesis.

The risk of bias among the included trials was largely low with 86.4% depicting low concerns. However, two studies had moderate risks (El-Khouly, 2021; Heiss, 2018) and one had a serious risk of bias (Bander, 2020). Based on the biases we reported in this study, future trials in this discipline of research ought to ensure datasets reporting of patient outcomes including follow-up. Furthermore, in an effort to improve outcomes for patients with DIPG, confounding variables including multi-disciplinary therapies must be assessed in a manner that may quantify various approaches both singularly or combined for survival (OS, EFS, PFS) and associated outcomes.

### 4.7. Limitations

There are certain limitations of this study. This study included early phase clinical trials that had a small sample size. We omitted case series and case reports to ensure only high-tier evidence was included, which may have led to omission of specific cases in literature. As such, the findings must be explored in further clinical trials to prove the safety, characterization of adverse events, and clinical outcomes if given in a larger sample size. While certain trials have demonstrated efficacy with prolonged survival rates, the results cannot be generalized and must be corroborated in further multi-center, blinded, randomized controlled trials to prove its benefits.

## 5. Conclusions

DIPG is a pediatric brain tumor that has a dismal prognosis with a median survival rate of nine months. There is currently no effective treatment beyond palliative radiotherapy. Our analysis of all clinical trials in the last five years captures 22 studies that point toward the direction in which the ongoing research is headed. Importantly, we show that a promising therapeutic strategy is a blood–brain barrier (BBB) bypass. We also find support for chemotherapeutic agents combined with VEGF- and EGFR inhibitors. Finally, our synthesis suggests re-irradiation as another strategy that can prolong survival in progressive DIPG patients. We collate current evidence to guide future research and therapeutic candidates in DIPG.

## Figures and Tables

**Figure 1 healthcare-11-00559-f001:**
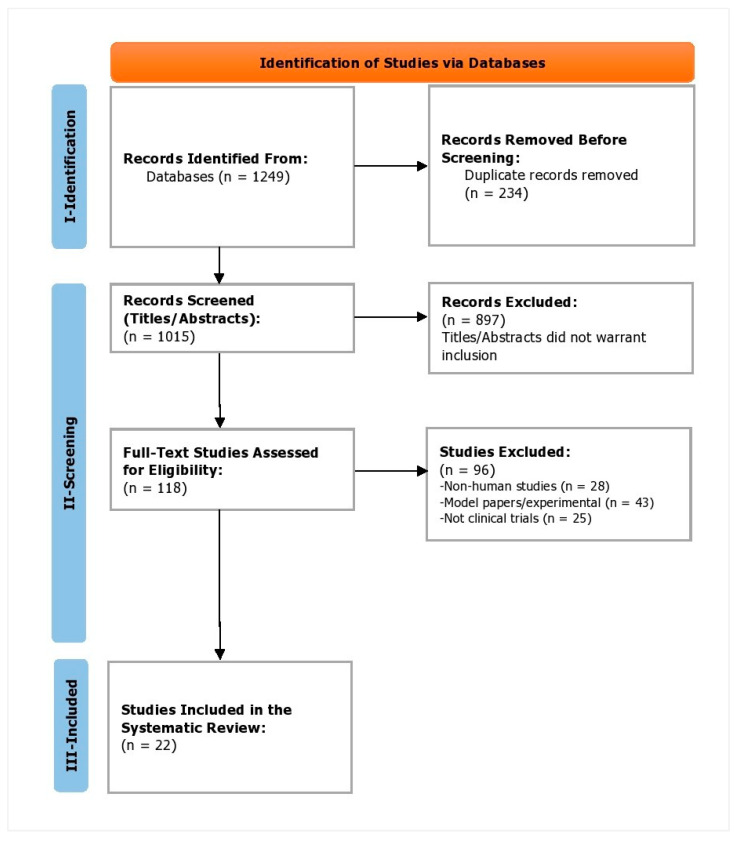
PRISMA flowchart depicting the study selection process.

**Figure 2 healthcare-11-00559-f002:**
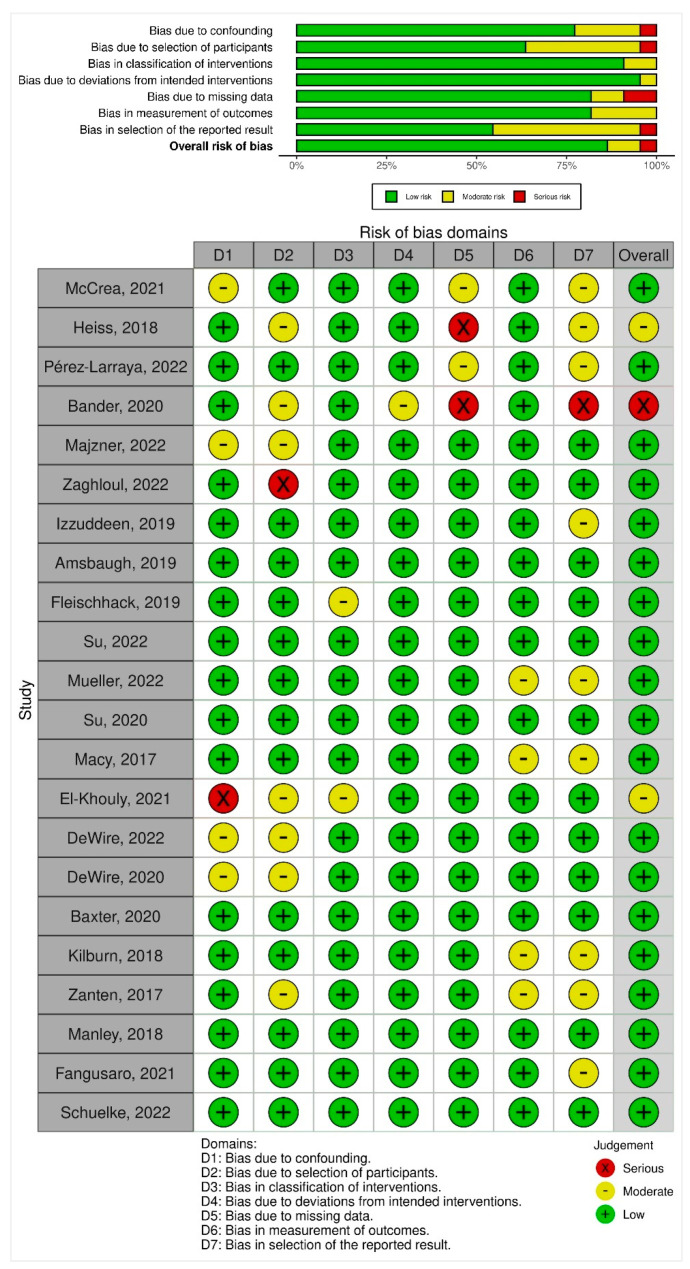
Summary plot and traffic light plot depicting risk of bias among the included studies [13,14,15,16,17,18,19,20,21,22,23,24,25,26,27,28,29,30,31,32,33,34].

**Table 1 healthcare-11-00559-t001:** Baseline characteristics of the trial and dosing regimens.

**Author**	**Year**	**Title**	**Journal**	**Country**	**Study Design**	**Inclusion Criteria**	**Intervention Given**	**Method of Administration**
*Blood-brain Barrier Bypass*								
McCrea	2021 [13]	Intraarterial delivery of bevacizumab and cetuximab utilizing blood-brain barrier disruption in children with high-grade glioma and diffuse intrinsic pontine glioma: results of a phase I trial	*Journal of Neurosurgery*	USA	Phase I trial (NCT01884740)	Patients aged <22 years with a histological diagnosis of relapsed or refractory HCG or radiological diagnosis of DIPG	Single intraarterial dose of 15 mg/kg bevacizumab and 200 mg/m^2^ cetuximab after BBBD with mannitol	Superselective intraarterial cerebral infusion
Heiss	2018 [14]	Phase I trial of convection-enhanced delivery of IL13-Pseudomonas toxin in children with diffuse intrinsic pontine glioma	*Journal of NeuroSurgery: Pediatrics*	USA	Phase I dose-escalation trial (NCT00088061)	Patients aged <18 years with clinical and radiological evidence of progressive DIPG, >2 weeks after their last chemotherapy dose or neurosurgical procedure, and >4 weeks from the last dose of radiation	IL13-PE38QQR and surrogate marker of IL13-PE38QQR distribution, Gd-DTPA co-infused, initial concentration of 0.125 μg/mL followed by 0.25 μg/mL IL13-PE38QQR	Intratumoral CED
Pérez-Larraya	2022 [15]	Oncolytic DNX-2401 Virus for Pediatric Diffuse Intrinsic Pontine Glioma	*The New England Journal of Medicine*	Spain	Phase 1 trial (NCT03178032)	Patients who were diagnosed with treatment-naive DIPG and aged 3–18 years	Single infusion of oncolytic adenovirus DNX-2401 (1 × 1010 or 5 × 1010 viral particles) followed by radiotherapy	CED, virus infused through a catheter placed in the cerebellar peduncle
Bander	2020 [16]	Repeat convection-enhanced delivery for diffuse intrinsic pontine glioma	*Journal of Neurosurgery*	USA	Phase I trial (NCT01502917)	Patients who were radiographically diagnosed with DIPG and previously treated with external-beam radiation therapy; showed no dose-limiting toxicity or disease progression in the 30-day observation window after first round of CED	≥2 infusions of I-8H9 monoclonal antibody (124I-omburtamab, Y-mAbs Therapeutics) via CED	CED through supratentorial trajectory with intraprocedural MRI-guided stereotactic placement
Majzner	2022 [17]	GD2-CAR T cell therapy for H3K27M-mutated diffuse midline gliomas	*Nature*	USA	Phase I dose-escalation trial (NCT04196413)	Patients who had H3K27M-mutated DIPG at any stage and aged 5–25 years	GD2-CAR T cells at dose level 1 (1 × 106 GD2-CAR T cells per kg administered intravenously) and subsequent GD2-CAR T-cell infusions administered intracerebroventricularly if there was a clinical benefit with first dose	Intravenously followed by intracerebroventricularly
**Author**	**Year**	**Title**	**Journal**	**Country**	**Study Design**	**Inclusion Criteria**	**Intervention Given**	**Method of Administration**
*Different Radiotherapy Regimens*								
Zaghloul	2022 [18]	Hypofractionated Radiation Therapy For Diffuse Intrinsic Pontine Glioma: A Noninferiority Randomized Study Including 253 Children	*International Journal of Radiation Oncology*	Egypt	Randomized clinical trial	Patients with diagnosed DIPG	HF radiotherapy regimens: HF1, receiving 39 Gy in 13 fractions; HF2, receiving 45 Gy in 15 fractions; and CF, receiving 54 Gy in 30 fractions	External beam radiotherapy
Izzuddeen	2019 [19]	Hypofractionated radiotherapy with temozolomide in diffuse intrinsic pontine gliomas: a randomized controlled trial	*Journal of Neuro-Oncology*	India	Phase II randomized trial	Patients aged 3 to 40 years with newly diagnosed patients with DIPG, confirmed radiologically on MRI with involvement of more than half of pons and basilar artery	Arm A received conventional fractionated RT of 60 Gy in 30 fractions over 6 weeks while patients in arm B received hypo-fractionated radiotherapy of 39 Gy in 13 fractions over 2.6 weeks along with concurrent TMZ 75 mg/m^2^ from day 1 to day 17 followed by adjuvant TMZ for six cycles.	Orally and external beam radiotherapy
Amsbaugh	2019 [20]	A Phase 1/2 Trial of Reirradiation for Diffuse Intrinsic Pontine Glioma	*International Journal of Radiation Oncology*	India	Phase I/II trial	Patients with radiologically confirmed DIPG by MRI who received radiation therapy at least 10 months previously with progressive disease confirmed clinically and radiologically	Re-irradiation at three doses, dose level 1: 24 Gy in 12 fractions, dose level 2: 26.4 Gy in 12 fractions, dose level 3: 30.8 Gy in 14 fractions	External beam radiotherapy
**Author**	**Year**	**Title**	**Journal**	**Country**	**Study Design**	**Inclusion Criteria**	**Intervention Given**	**Method of Administration**
*Non-Chemotherapeutic Agent Regimens*								
Fleischhack	2019 [21]	Nimotuzumab and radiotherapy for treatment of newly diagnosed diffuse intrinsic pontine glioma (DIPG): a phase III clinical study	*Journal of Neuro-Oncology*	Germany, Italy, Russia	Phase III trial	Patients between 3 and 20 years with clinically and radiologically confirmed DIPG, diagnosed in the last 3 months	Nimotuzumab was given intravenously at 150 mg/m^2^ weekly for 12 weeks and radiotherapy at a total dose of 54 Gy between week 3 and week 9	Intravenously and external beam radiotherapy
Su	2022 [22]	Phase I/II trial of vorinostat and radiation and maintenance vorinostat in children with diffuse intrinsic pontine glioma: A Children’s Oncology Group report	*Neuro-Oncology*	USA	Phase I/II trial	Children aged 3 to 21 years with newly diagnosed DIPG, radiographically defined as tumors with a pontine epicenter and diffuse involvement of at least two-thirds of the pons; if aforementioned criteria were not met then tumors were biopsied, children who had anaplastic astrocytoma, glioblastoma, gliosarcoma, or anaplastic mixed glioma were included	Vorinostat once daily from Monday till Friday, during radiation therapy (54 Gy in 30 fractions), then 230 mg/m^2^ daily for a maximum of twelve 28-day cycles	Orally and external beam radiotherapy
Mueller	2022 [23]	Wee1 kinase inhibitor adavosertib with radiation in newly diagnosed diffuse intrinsic pontine glioma: A Children’s Oncology Group phase I consortium study	*Neuro-Oncology Advances*	USA	Phase I trial (NCT01922076)	Patients aged 3–21 years with newly diagnosed DIPG	On days of cranial radiation therapy, 7 adavosertib DLs (50 mg/m^2^ alternating weeks, 50 mg/m^2^ alternating with weeks of every other day, 50 mg/m^2^, then 95, 130, 160, 200 mg/m^2^)	Orally
Su	2020 [24]	A phase 2 study of valproic acid and radiation, followed by maintenance valproic acid and bevacizumab in children with newly diagnosed diffuse intrinsic pontine glioma or high-grade glioma	*Pediatric Blood & Cancer*	USA	Phase II trial (NCT00879437)	Patients between 3 and 21 years with newly diagnosed radiologically confirmed DIPG	Radiation therapy and VPA (15 mg/kg/day and dose adjustment to maintain a trough range of 85 to 115 μg/mL), VPA continued post-radiation, bevacizumab (10 mg/kg) intravenously biweekly, four weeks after completing radiation therapy	Intravenously and external beam radiotherapy
**Author**	**Year**	**Title**	**Journal**	**Country**	**Study Design**	**Inclusion Criteria**	**Intervention Given**	**Method of Administration**
*Chemotherapeutic Agent Regimens*								
Macy	2017 [25]	A pediatric trial of radiation/cetuximab followed by irinotecan/cetuximab in newly diagnosed diffuse pontine gliomas and high-grade astrocytomas: A Pediatric Oncology Experimental Therapeutics Investigators’ Consortium study	*Pediatric Blood & Cancer*	USA	Phase II trial	Patients 3 to 21 years of age with newly diagnosed DIPG confirmed clinically and radiologically (MRI)	EBT (5940 cGy in 33 fractions (180 cGy)) with concomitant cetuximab (250 mg/m^2^ IV weekly for six doses) with a recovery period of 26–52 days and followed by irinotecan (16 mg/m^2^/day over one hour for five days, given two consecutive weeks) and cetuximab once weekly (250 mg/m^2^/dose IV in 21-day cycles)	Intravenously and external beam radiotherapy
El-Khouly	2021 [26]	A phase I/II study of bevacizumab, irinotecan, and erlotinib in children with progressive diffuse intrinsic pontine glioma	*Journal of Neuro-Oncology*	The Netherlands	Phase I/II trial (EudraCT 2009-016080-11, Dutch Trial Register NTR2391)	Patients aged 3–18 years with progressive DIPG (clinical or radiological) after initial radiotherapy	Biweekly bevacizumab (10 mg/kg) and irinotecan (125 mg/m^2^) combined with daily erlotinib, dose increased for erlotinib for 2 cohorts (65 and 85 mg/m^2^) following a 3 + 3 dose-escalation schedule, until disease progression for a maximum of one year	Through central venous catheter and intravenously
DeWire	2022 [27]	Phase I study of ribociclib and everolimus in children with newly diagnosed DIPG and high-grade glioma: A CONNECT pediatric neuro-oncology consortium report	*Neuro-Oncology Advances*	USA	Phase I trial (NCT02607124)	Patients with newly diagnosed DIPG or HIG who initiated radiotherapy within 30 days of radiographic diagnosis or definitive surgery (whichever was later)	Ribociclib 170 mg/m^2^ daily for 21 days and everolimus 1.5 mg/m^2^ daily for 28 days	Orally, via g-tube, or nasogastric tube
DeWire	2020 [28]	A phase I/II study of ribociclib following radiation therapy in children with newly diagnosed diffuse intrinsic pontine glioma (DIPG)	*Journal of Neuro-Oncology*	USA	Phase I/II trial (NCT02607124)	Patients aged 1–30 years with newly-diagnosed DIPG confirmed radiologically or histologically, within 30 days of radiographic diagnosis or definitive surgery	350 mg/m^2^ ribociclib daily for 21 days/7 days of every 28 days for up to 12 courses, 2–4 weeks after radiotherapy	Orally or via nasogastric/gastric tube
Baxter	2020 [29]	A phase I/II study of veliparib (ABT-888) with radiation and temozolomide in newly diagnosed diffuse pontine glioma: a Pediatric Brain Tumor Consortium study	*Neuro-Oncology*	USA	Phase I/II trial (NCT01514201)	Children aged 21 years or younger with newly diagnosed DIPG, defined as tumors with a pontine epicenter and diffuse intrinsic involvement of the pons	Veliparib was given Monday through Friday (50 mg/m^2^/dose twice daily with 2 planned dose escalations, 65 and 85 mg/m^2^/dose twice daily and 1 planned de-escalation (35 mg/m^2^/dose twice daily) during radiation (5400 cGy in 30 fractions over 6 weeks) and a 4-week gap, followed by veliparib at 25 mg/m^2^ b.i.d. and TMZ 135 mg/m^2^ daily for 5 days every 28 days	Orally
Kilburn	2018 [30]	A pediatric brain tumor consortium phase II trial of capecitabine rapidly disintegrating tablets with concomitant radiation therapy in children with newly diagnosed diffuse intrinsic pontine gliomas	*Pediatric Blood & Cancer*	USA	Phase II trial	Children aged 3 to 17 years with newly diagnosed DIPG	Capecitabine, 650 mg/m^2^/dose BID (MTD in children with concurrent radiation) was administered for 9 weeks starting the first day of RT. Following a 2-week break, 3 courses of capecitabine, 1250 mg/m^2^/dose BID for 14 days followed by a 7-day rest, were administered	Orally and external beam radiation
Zanten	2017 [31]	A phase I/II study of gemcitabine during radiotherapy in children with newly diagnosed diffuse intrinsic pontine glioma	*Journal of Neuro-Oncology*	The Netherlands	Phase I/II trial	Patients with newly-diagnosed DIPG	Gemcitabine (weekly dose for 6 weeks, increasing doses of 140, 175, and 200 mg/m^2^ gemcitabine, respectively, following a 3 + 3 dose-escalation schedule) concomitant to 6 weeks of hyper fractionated radiotherapy	Intravenously and external beam radiation
Manley	2018 [32]	A phase 1/2 dose-finding, safety, and activity study of cabazitaxel in pediatric patients with refractory solid tumors including tumors of the central nervous system	*Pediatric Blood & Cancer*	USA	Phase I/II dose-escalating trial (NCT01751308)	Patients aged 2–18 years old with progressive DIPG and recovered from any acute toxic effects of all prior therapy to grade ≤1 before enrollment	Cabazitaxel infused over 1 h (20 mg/m^2^ initially and if tolerated, escalated to 25, 30, 35, and 40 mg/m^2^) on day 1 of every 21-day cycle	Intravenously
**Author**	**Year**	**Title**	**Journal**	**Country**	**Study design**	**Inclusion criteria**	**Intervention given**	**Method of administration**
*Immunotherapy*								
Fangusaro	2021 [33]	Phase 2 Study of Pomalidomide (CC-4047) Monotherapy for Children and Young Adults With Recurrent or Progressive Primary Brain Tumors	*Frontiers in Oncology*	USA, France, Italy, Spain, UK	Phase II trial (NCT03257631)	Patients aged 1 to <21 years with a diagnosis of recurrent or progressive DIPG, must have received ≥1 prior standard therapy	Pomalidomide (2.6 mg/m^2^/day once daily) on days 1–21 of each 28-day treatment cycle, followed by a 7-day rest period for up to 24 cycles	Orally
Schuelke	2022 [34]	Phase I trial of sargramostim/pelareorep therapy in pediatric patients with recurrent or refractory high-grade brain tumors	*Neuro-Oncology Advances*	USA	Phase I trial (NCT02444546)	Patients aged 10 to 21 years with progressive high-grade DPIG and life expectancy >3 months	Sargramostim (subcutaneously at 250 mcg/m^2^) for 2 days followed by 3 days of pelareorep (IV over 60 min)	Intravenously and subcutaneously

Abbreviations: BBBD: blood–brain barrier disruption; BID: two times a day; CED: convection-enhanced delivery; CF: conventional fractionation; DLs: dose levels; Gd-DTPA: gadolinium-diethylenetriamine-pentaacetic acid; Gy: gray; HCG: high-grade glioma; HF: hypofractioned; MRI: magnetic resonance imaging; MTD: maximum tolerated dose; TMZ: temozolomide; VPA: valproic acid.

**Table 2 healthcare-11-00559-t002:** Patient characteristics and outcome measures.

**Author**	**Number of Patients with DIPG**	**Age at Diagnosis (Years)**	**Sex (% Male)**	**Previous Treatment**	**Outcome Measures**
*Blood–Brain Barrier Bypass*					
McCrea	10 patients	5.5 years (5–7)	5 patients (50%)	All patients had received standard radiotherapy, 6 patients (60%) had received immunotherapy, convection-enhanced delivery, MK-1775/Wee1 inhibitor, and oral panobinostat	Clinical response, safety, objective response (T1-weighted pre- and postcontrast sequences and T2-weighted FLAIR sequences)
Heiss	5 patients	Mean: 13 years (SD: 5)	3 patients (60%)	Standard radiotherapy	Clinical response, radiological response, corticosteroid dose, QoL
Pérez-Larraya	12 patients	NR	NR	None	Safety, overall survival, quality of life, objective response, tumor biopsy and peripheral-blood samples for correlative studies of the molecular features of DIPG and antitumor immune responses
Bander	7 patients	Mean: 5.4 years	4 patients (57.14%)	All patients received external-beam RT	Postinfusion deficits, distribution volume, targeting accuracy
Majzner	3 patients	Mean: 13.3 years	1 patient (33.3%)	All patients received standard radiotherapy ≥ 6 months before enrollment	Safety, clinical improvement, radiological improvement
**Author**	**Number of Patients with DIPG**	**Age at Diagnosis (Years)**	**Sex (% Male)**	**Previous Treatment**	**Outcome Measures**
*Different Radiotherapy Regimens*					
Zaghloul	253 patients	NR	NR	NR	OS, PFS
Izzuddeen	33 patients (conventional treatment arm: *n* = 16, experimental arm: *n* = 17)	<7 years: conventional arm: 9 patients (52%), experimental arm: 9 patients (50%); 8–18 years: conventional arm: 5 patients (29%), experimental arm: 5 patients (27%); >18 years: conventional arm: 3 patients (17%), experimental arm: 4 patients (22%)	Conventional arm: 8 patients (47%), treatment arm: 7 patients (38%)	None	Toxicities, PFS, OS
Amsbaugh	12 patients (group 1: *n* = 6, group 2: *n* = 4, group 3: *n* = 2)	Group 1: 5.5 years (4–20), group 2: 10 years (5–26) group 3: 6 years (5–7)	7 patients (58.3%)	Radiotherapy (at least 10 months before)	Toxicities, OS, PFS, clinical improvement, radiological response, QoL
**Author**	**Number of Patients with DIPG**	**Age at Diagnosis (Years)**	**Sex (% Male)**	**Previous Treatment**	**Outcome Measures**
*Non-Chemotherapeutic Agent Regimens*					
Su	61 patients	7.1 years (3.3–19.4)	32 patients (45.7%)	None	Toxicities, EFS, OS
Fleischhack	42 patients	Median: 7.4 years (3–15)	16 patients (38.1%)	None	PFS, clinical response, radiological response (RECIST criteria), adverse events
Mueller	46 patients	Median: 6 years (3–21)	22 patients (48%)	None (except surgery)	Tolerability, pharmacokinetics, OS, radiological response, peripheral blood γH2AX levels
Su	20 patients	Median: 7.69 years (5.2–9.9)	10 patients (50%)	May have received surgery and/or corticosteroids	Safety, radiological response (WHO bidimensional criteria), EFS, OS
**Author**	**Number of Patients with DIPG**	**Age at Diagnosis (Years)**	**Sex (% Male)**	**Previous Treatment**	**Outcome Measures**
*Chemotherapeutic Agent Regimens*					
Macy	25 patients	Median: 8 years (3–19)	21 patients (47%)	None (except surgery)	Tolerability, OS, PFS, TTP
El-Khouly	9 patients	Mean: 9.39 years	5 patients (55.5%)	Radiotherapy (all 9 patients, 100%) combined with gemcitabine (4 patients, 44.4%) in the previous phase I of the trial, re-irradiation during the current trial (4 patients, 44.4%)	Safety (DLTs), sPFS, OS, radiological response (MRI images scored with the modified RANO-criteria), QoL
DeWire	15 patients	Median: 6.5 years (2–15)	5 patients (26%)	Patients were eligible if they received 10% of standard dose of radiotherapy (54 Gy across 1.8 Gy daily fractions over 6 weeks to the planning target volume)	Toxicities, OS, radiological response (MRI based on RANO criteria)
DeWire	9 patients	7.3 years (5–14.7)	4 patients (40%)	Patients were eligible if they received 10% of standard dose of radiotherapy (54 Gy across 1.8 Gy daily fractions over 6 weeks to the planning target volume)	Safety, OS, feasibility
Baxter	65 patients	6.6 years (2.2–15.8)	40 (61.5%)	None	OS, radiological response, DLT
Kilburn	44 patients	7.2 (3.4–16.2)	22 (50%)	None (except surgery and corticosteroid therapy)	PFS, safety
Zanten	9 patients	10.8 years (7.5–17.3)	5 patients (55.5%)	None	DLTs, radiological response (based on modified RANO criteria), PFS, OS, QoL
Manley	12 patients	Phase 1: 9.0 years (4–18), phase 2: 9.5 years (3–16)	Phase 1: 15 patients (65%), phase 2: 8 patients (50%)	Systemic anticancer therapy within ≤3 weeks, investigational agents, or small field radiotherapy ≤4 weeks, craniospinal radiation therapy ≤6 months	Radiological response (as per the modified RANO criteria), PFS, DLTs
**Author**	**Number of Patients with DIPG**	**Age at Diagnosis (Years)**	**Sex (% Male)**	**Previous Treatment**	**Outcome Measures**
*Immunotherapy*					
Fangusaro	9 patients	7.0 (4–12)	7 (63.6%)	Radiation (11 patients, 100%), surgery (5 patients, 55.55%), systemic therapy (7 patients, 63.6%)	OR, LTSD, OS, PFS, safety
Schuelke	2 patients	10 and 17 years	0	Radiation (2 patients, 100%), chemotherapy (2 patients, 100%)	DLTs, radiological response, OS

Abbreviations: DLTs: dose limiting toxicities; EFS: event free survival; LTSD: long-term stable disease; MRI: magnetic resonance imaging; NR: not reported; OS: overall survival; PFS: progression-free survival; QoL: quality of life; RT: radiotherapy; sPFS: secondary progression-free survival; TTP: time to progression; WHO: World Health Organization.

**Table 3 healthcare-11-00559-t003:** Efficacy and safety outcomes of the trials.

**Author**	**Median OS (Months)**	**Median EFS/PFS (Months)**	**Radiological Response (%)**	**Clinical Improvement (*n*, %)**	**Tolerance and Safety**	**Steroid Use Discontinuation**	**Concluding Remarks**
*Blood–Brain Barrier Bypass*							
McCrea	17.3 months (221–761 days)	NR	T1-weighted postcontrast sequence: progressive disease (4 patients, 40%), stable disease (2 patients, 20%), partial response (2 patients, 20%), complete response (1 patient, 10%); T2-weighted FLAIR imaging: progressive disease (5 patients, 50%), stable disease (5 patients, 50%)	6 patients (60%)	Well-tolerated; 4 (40%) patients had minor adverse events (grade I epistaxis in 2 patients and grade I rash in 2 patients)	2 patients (20%)	Intraarterial therapy of bevacizumab and cetuximab is well-tolerated in children with DIPG and warrants further investigation
Heiss	NR	NR	Disease progression: 3 patients (60%)	1 patient (20%)	Elevated serum creatine kinase (2 patients, 40%), renal calculi (1 patient, 20%), somnolence (1 patient, 20%), suspected aspiration prompting hospitalization (1 patient, 20%)	4 patients (80%)	Direct brainstem infusion of IL13-PE using CED temporarily arrested disease progression in 2 out of 5 patients and adverse events were due to infusion-related brainstem edema with no signs of toxicity noted
Pérez-Larraya	17.8 months (5.9–33.5)	NR	MRI: complete response (9 patients, 75%) partial response (3 patients, 25%), stable disease (8 patients, 66.7%)	NR	Grade 1 and 2 events: headache, nausea, vomiting, fatigue, hemiparesis (1 patient, 8.3%), tetraparesis (1 patient, 8.3%)	NR	Intratumoral infusion of oncolytic virus DNX-2401 followed by radiotherapy in pediatric patients with DIPG has molecular changes including T-cell activity and a reduction in or stabilization of tumor size but there may be adverse events
Bander	NR	NR	NR	NR	Grade 1 and 2 events: contralateral hemiparesis (5 patients, 71.4%), nystagmus (2 patients, 28.6%), dysmetria (1 patient, 14.3%), cranial nerve VI and/or VII palsy (4 patients, 51.1%)	NR	Repeated CED in the brainstem for children with DIPG is safe
Majzner	Patient 1: 13 months, Patient 2: 20 months, Patient 3: 26 months	NR	MRI: 20% enlargement (1 patient, 33.3%), improved T2/FLAIR signal (2 patients, 66.7%), 17% smaller tumor volume (1 patient, 33.3%)	2 patients (66.7%)	CAR T cell-mediated inflammation at the local tumor site, termed TIAN in all 3 patients (100%)	3 patients (100%)	Toxicity management algorithm for TIAN offers potentially safe delivery of targeted CAR T-cell therapy locally
**Author**	**Median OS (Months)**	**Median EFS/PFS (Months)**	**Radiological Response (%)**	**Clinical Improvement (*n*, %)**	**Tolerance and Safety**	**Steroid Use Discontinuation**	**Concluding Remarks**
*Different Radiotherapy Regimens*							
Zaghloul	HF1: 9.6 months, HF2: 8.2 months, CF: 8.7 months	NR	NR	NR	Well-tolerated	NR	Hypofractioned radiation therapy is non-inferior to conventional fractionation with younger age (2–5 years) showing superiority with HF1 (low hypofractionated therapy dose of 39 Gy in 13 fractions)
Izzuddeen	Conventional arm: 11 months (95% CI: 7.5–14.5), experimental arm: 12 months (95% CI: 10.5–13.5)	PFS: conventional arm—7 months (95% CI: 3.6–10.3), experimental arm—8 months (95% CI: 6.7–9.3)	NR	NR	5 patients (28%) in the experimental arm developed ≥ grade 3 hematological toxicity; 1 patient (7%) developed ≥ grade 3 toxicity	NR	Hypofractionated radiotherapy with concurrent and adjuvant temozolomide did not improve survival rates and has higher hematological toxicity
Amsbaugh	19.5 months (95% CI: 15.6–21.1)	PFS: 4.5 months	Improvement: 8 patients (66.7%)	11 (91.7%)	Dose level 3: ≥Grade 3 events: hypoxia and dysphagia (1 patient, 50%)	NR	Re-irradiation was safe and had improved survival outcomes among patients with progressive DIPG
**Author**	**Median OS (Months)**	**Median EFS/PFS (Months)**	**Radiological Response (%)**	**Clinical Improvement (*n*, %)**	**Tolerance and Safety**	**Steroid Use Discontinuation**	**Concluding Remarks**
*Non-Chemotherapeutic Agent Regimens*							
Su	1-year OS: 39.2% (95% CI: 27.8–50.5%).	1-year EFS: 5.85% (95% CI 1.89–13.1%)	NR	NR	42 patients (60%) had at least 1 DLT	NR	Vorinostat given together with radiation and afterward was well-tolerated but did not improve survival outcomes
Fleischhack	9.4 months	PFS: 5.8 months	ORR: 4.2%, Partial response (2 patients, 4.8%), stable disease (27 patients, 64.3%), progressive disease (10 patients, 23.8%)	NR	Alopecia (6 patients, 14.3%), vomiting (3 patients, 7.1%), headache (3 patients, 7.1%) radiation skin injury (3 patients, 7.1%), intra-tumoral bleeding (1 patient, 2.4%), acute respiratory failure (1 patient, 2.4%)	NR	Nimotuzumab combined with RT is well-tolerated and has comparable efficacy with RT and intensive chemotherapy without requiring prolonged hospitalization among children with newly diagnosed DIPG
Mueller	11.8 months (9–13.9)	NR	Stable disease: 33 patients (80.5%), progressive disease: 8 patients (19.5%)	NR	≥Grade 3 events: ALT elevation (1 patient, 6.7%), neutropenia (1 patient, 6.7%)	NR	Adavosertib in combination with CRT is well tolerated in children with newly diagnosed DIPG, however, compared to historical controls, did not improve OS. These results can inform future trial designs in children with high-risk cancer.
Su	10.3 (7.4–13.4) months	EFS: 7.8 (95% CI 5.6–8.2)	Partial response (8 patients, 40%), minor response (9 patients, 45%), stable disease (1 patient, 5%), not available (2 patients, 10%)	NR	≥Grade 3 events with VPA and RT: thrombocytopenia (3 patients), somnolence (1 patient), fatigue (3 patients), weight gain (2 patients); ≥grade 3 events with VPA and bevacizumab maintenance: thrombocytopenia (3 patients), intracranial/intratumoral hemorrhage (1 patient), hypertension (4 patients), subacute bone infarction (1 patient), fatigue (3 patients), weight gain (2 patients)	NR	VPA and bevacizumab given in combination with radiation is well-tolerated but there is no improvement in EFS or OS among children with newly diagnosed DIPG
**Author**	**Median OS (Months)**	**Median EFS/PFS (Months)**	**Radiological Response (%)**	**Clinical Improvement (*n*, %)**	**Tolerance and Safety**	**Steroid Use Discontinuation**	**Concluding Remarks**
*Chemotherapeutic Agent Regimens*							
Macy	12.1 months (95% CI: 9.93, 18)	PFS: 7.12 months (95% CI: 6.89, 12.5)	Stable disease: 6 patients (24%)	NR	≥Grade 3 events: lymphopenia (26 patients, 57.7%), hypokalemia (18 patients, 40%), neutropenia (6 patients, 13.3%), anorexia (9 patients, 20%)	NR	Combination of EGFR inhibitor to radiation and irinotecan is a treatment regimen that may improve progression-free survival and is well-tolerated but did not improve survival rates
El-Khouly	Overall: 13.8 months (9.3–33.0), patients who were re-irradiated: 16.2 months (12.8–20.0)	PFS: 7.3 months (3.5–10.0), sPFS: 3.2 months (1.0–10.9)	3 months: Partial response (3 patients, 33.3%), stable disease (1 patient, 11.1%), progressive disease (5 patients, 55.5%); 6 months: progressive disease (2 patients, 50%). stable disease (2 patients, 50%)	4 patients (44.4%)	Grade III acute diarrhea (1 patient, 11/1%), grade II acute secretory diarrhea (1 patient, 11.1%). grade I/II late-onset diarrhea (5 patients, 55.5%), grade I/II nausea and vomiting (4 patients, 44.4%), grade I acneiform rash (5 patients, 55.5%), grade I/II mucositis (1 patient, 11.1%), grade I/II constipation (1 patient, 11.1%), grade II keratitis (1 patient, 11.1%), grade II urinary tract infection (2 patients, 22.2%), and grade II adrenal insufficiency as a result of chronic dexamethasone use (2 patients, 22.2%)	NR	Daily erlotinib (up to 85 mg/m^2^) with biweekly bevacizumab and irinotecan is safe and improves median OS in children with progressive DIPG
DeWire	13.9 months	NR	Pseudoprogression: 4 patients (8.3%), progression: 2 patients (4.2%)	NR	≥Grade 3 events: neutropenia (6 patients, 33%), leucopenia (3 patients, 17%), lymphopenia (2 patients, 11%), pulmonary infection (1 patient, 6%), elevated ALT and hypokalemia (1 patient, 6%), cardiac toxicity (1 patient, 6%)	NR	Ribociclib and everolimus following radiotherapy in children with newly diagnosed DIPG is well-tolerated and requires further exploration for efficacy potential
DeWire	16.1 months (10–30)	NR	Disease progression: 9 patients (90%)	NR	≥grade 3 events: neutropenia (9 patients, 90%), lymphopenia (5 patients, 50%), and leukopenia (7 patients, 70%)	NR	Ribociclib administered following radiotherapy has survival benefits but increased tumor necrosis may be a treatment effect represent a treatment effect
Baxter	One-year OS: 37.2% (SE 7%)	NR	PR: 7 patients (14%)	NR	DLTs: 4 patients (33.3%)during radiation in phase I (Grade 2 intratumoral hemorrhage (*n* = 1), grade 3 maculopapular rash (*n* = 2), and grade 3 nervous system disorder (generalized neurologic deterioration) (*n* = 1)), 4 patients (50%) during intrapatient dose escalation	NR	Veliparib used in combination with radiation followed by TMZ and veliparib was well-tolerated and did not improve survival rates in patients with newly diagnosed DIPG
Kilburn	NR	6-month PFS: 33.7% (SE = 7.1%), 1-year PFS: 7.2% (SE = 3.5%)	Progressive disease (8 patients, 18.2%)	Deterioration (3 patients, 6.8%)	DLTs: 5 patients (grade 4 neutropenia (*n* = 1), grade 2 CNS necrosis (*n* = 2), grade 4 neutropenia that did not resolve within 7 days (*n* = 1), and persistent toe infection (*n* = 1)	NR	Concomitant and adjuvant Capecitabine with radiotherapy was well-tolerated but did not improve survival outcomes for children with newly-diagnosed DIPG
Zanten	Intermediate risk: 12.4 months, high risk: 8.1 months	PFS for intermediate risk: 6.4 months, PFS for high risk: 4.5 months	Stable disease: 2 patients, progressive disease: 3 patients, pseudoprogression: 4 patients	9 patients (100%)	Grade 3 hepatotoxicity (2 patients, 22.2%), grade 3 neutropenia (1 patient, 11.1%)	3 patients (75%)	Gemcitabine in combination with radiotherapy is well-tolerated but does not improve survival outcomes in patients with newly-diagnosed DIPG
Manley	2.7 months (95% CI: 1.7–4.5)	Median PFS: 1.3 months (95% CI: 0.6–2.1)	Progressive disease 25 patients (75.8%), stable disease: 6 patients (18.2%), partial response: 1 patient (3%), complete response: 1 patient (3%)	NR	≥grade 3 events: 12 patients (52%)	Steroid treatment used as part of protocol	Cabazitaxel in pediatric patients with progressive DIPG does not improve survival outcomes but is well-tolerated and safe at the established MTD
**Author**	**Median OS (Months)**	**Median EFS/PFS (Months)**	**Radiological Response (%)**	**Clinical Improvement (*n*, %)**	**Tolerance and Safety**	**Steroid Use Discontinuation**	**Concluding Remarks**
*Immunotherapy*							
Fangusaro	3.78 months	Median PFS: 2.6 months	Disease progression: 6 (66.7%)	NR	≥grade 3 events: neutropenia (3 patients, 27.3%), lymphopenia (1 patient, 9.1%), leucopenia (1 patient, 9.1%), vertigo (1 patient, 9.1%)	NR	Pomalidomide monotherapy in progressive DIPG did not improve survival rates
Schuelke	1.1 months	NR	Disease progression: 2 patients (100%)	Death: 82 and 60 days	No DLTs	NR	Sargramostim/pelareorep was well-tolerated but did not improve survival rates

Abbreviations: ALT: alanine transaminase; CAR: chimeric antigen receptor; CED: convection-enhanced delivery; CI: confidence interval; EFS: event-free survival; EGFR: epidermal growth factor receptor; MTD: maximum tolerated dose; NR = not reported; PFS: progression free survival; RT: radiotherapy; SE: standard error; TIAN: tumor inflammation-associated neurotoxicity; VPA: valproic acid.

## Data Availability

All data utilized for the purpose of this study are available publicly and online. Additional data may be requested by the corresponding author (A.S.) on reasonable request.

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
