# Peer review of "Emerging Therapeutic Strategies for Diffuse Intrinsic Pontine Glioma: A Systematic Review"

_healthcare, 2023, doi:10.3390/healthcare11040559_

Round 1

Reviewer 1 Report

The authors conducted a thorough review of the literature of the last five years to select clinical trials about diffuse intrinsic pontine gliomas (DIPG). DIPG affect mostly young people and have a dismal prognosis. The final analysis was conducted on 22 papers with a total of 703 patients enrolled in clinical trials. While palliative radiotherapy is still the standard treatment, efforts have been made toward chemotherapy bypassing the blood-brain barrier added to molecular targeted therapy with agents against VEGF and EGFR.

From the technical point of view, the study has been well planned and conducted rigorously: the results constitute a useful and reliable update on the matter.

Author Response

Before you read my responses to your comments, I humbly thank you very much for your time and due diligence in reviewing the paper. The voluntary nature of work must be acknowledged.

Regards,

Zouina S.

Reviewer 1 Comments and Author Responses:

Comment 1: The authors conducted a thorough review of the literature of the last five years to select clinical trials about diffuse intrinsic pontine gliomas (DIPG). DIPG affect mostly young people and have a dismal prognosis. The final analysis was conducted on 22 papers with a total of 703 patients enrolled in clinical trials. While palliative radiotherapy is still the standard treatment, efforts have been made toward chemotherapy bypassing the blood-brain barrier added to molecular targeted therapy with agents against VEGF and EGFR.

Response to Comment 1: Thank you very much for reviewing our important study and providing feedback. It is greatly appreciated.

Comment 2: From the technical point of view, the study has been well planned and conducted rigorously: the results constitute a useful and reliable update on the matter.

Response to Comment 2: Thank you very much for reviewing our important study and providing feedback. It is greatly appreciated.

Reviewer 2 Report

The diagnosis of a diffuse intrinsic pontine glioma (DIPG) is nowadays still a challenging situation for a (pediatric) oncologist. The prognosis is dismal, and the care is more palliative than curative. The median survival rate is around one year. So it is always difficult to tell the parents that there are no other curative treatment options after searching for more current treatment strategies.

This systematic review aims to collate recent clinical trial data for diffuse intrinsic pontine glioma (DIPG) and show the most promising therapies of the current five years. Both adult and pediatric patients with newly diagnosed or progressive DIPG were considered in the clinical trial setting. The risk of bias was assessed using the ROBINS-I tool. A total of 22 trials were included reporting the efficacy and safety outcomes among patients. The synthesis suggests that reirradiation may prolong survival in patients with progressive DIPG; it also instills that palliative radiotherapy has been a critical prognostic choice for patients.

The article is well-organized and contains all the necessary components. The sections are well-developed. The author did an excellent job synthesizing the literature, which is helpful indeed. In addition, they answer the questions they set out to answer.

The methodology is clearly explained: PubMed/MEDLINE, Web of Science, Scopus, and Cochrane were systematically searched using the correct keywords. As a result, the article is well-written and easy to understand.

The author's results can convince the readers.

There are some questions and comments.

Theoretically, case series and case reports were excluded; however, some trials with 2 or 3 participants have been included.

Table 2 is too long; structuring it, perhaps repeating the header, would make it more transparent. In addition, the inclusion of abbreviations (AR, AN) would be necessary.

Figure 2. summarizes the risk of bias among the included studies. It would be helpful to write some explanation remarks in the discussion at the detailed study description about the bias.

There is a repetition at the end of the discussion:

"There are certain limitations. There are certain limitations of this study." 

Author Response

Before you read my responses to your comments, I humbly thank you very much for your time and due diligence in reviewing the paper. The voluntary nature of work must be acknowledged.

Regards,

Zouina S.

Reviewer 2 Comments and Author Responses:

Comment 1: The diagnosis of a diffuse intrinsic pontine glioma (DIPG) is nowadays still a challenging situation for a (pediatric) oncologist. The prognosis is dismal, and the care is more palliative than curative. The median survival rate is around one year. So it is always difficult to tell the parents that there are no other curative treatment options after searching for more current treatment strategies.

Response to Comment 1: Thank you for providing your valuable insight to the subject matter. I fully agree with it.

Comment 2: This systematic review aims to collate recent clinical trial data for diffuse intrinsic pontine glioma (DIPG) and show the most promising therapies of the current five years. Both adult and pediatric patients with newly diagnosed or progressive DIPG were considered in the clinical trial setting. The risk of bias was assessed using the ROBINS-I tool. A total of 22 trials were included reporting the efficacy and safety outcomes among patients. The synthesis suggests that reirradiation may prolong survival in patients with progressive DIPG; it also instills that palliative radiotherapy has been a critical prognostic choice for patients.

Response to Comment 2: Thank you once again for critically reviewing my paper. I believe the paper is the need of the hour particularly because some breakthroughs may help improve survival in patient populations. 

Comment 3: The article is well-organized and contains all the necessary components. The sections are well-developed. The author did an excellent job synthesizing the literature, which is helpful indeed. In addition, they answer the questions they set out to answer.

Response to Comment 3: Thank you for your constructive criticism. 

Comment 4: The methodology is clearly explained: PubMed/MEDLINE, Web of Science, Scopus, and Cochrane were systematically searched using the correct keywords. As a result, the article is well-written and easy to understand.The author's results can convince the readers.

Response to Comment 4: Thank you for identifying the strengths of my paper.

There are some questions and comments.

Comment 5: Theoretically, case series and case reports were excluded; however, some trials with 2 or 3 participants have been included.

Response to Comment 5: While the authors sought to ensure rigorous studies only in this study, we contemplated including the series and reports. However, the regulatory and other checks that are employed in trials are oftentimes overlooked in reports/series. Therefore, if laboratory/data scientists were to read our paper, which I anticipate they will, I only wish to include data sources that are typically considered of top-tier quality. Hence, omissions of case series and case reports was the final verdict among the delphi processing phase. I will go on to add this as a limitation of our paper to ensure full transparency.

Comment 6: Table 2 is too long; structuring it, perhaps repeating the header, would make it more transparent. In addition, the inclusion of abbreviations (AR, AN) would be necessary.

Response to Comment 6: I have taken your consideration well. I have restructured the tables. Highlighted headings. Included headers before new subsections. Additionally, have also included abbreviations. The page count for the tables has been reduced by 4 pages. Moreover, to reduce and make the table flow better, I have added abbreviations as required and listed them at the bottom of the table for easy reviewing.

Comment 7: Figure 2. summarizes the risk of bias among the included studies. It would be helpful to write some explanation remarks in the discussion at the detailed study description about the bias.

Response to Comment 7: I have gone on to add more specific details in a manner that will help guide future researchers in this area; Excerpt as follows: “The risk of bias among the included trials was largely low with 86.4% depicting low concerns. However, 2 studies had moderate risks (El-Khouly, 2021; Heiss, 2018) and one had a serious risk of bias (Bander, 2020). Based on the biases we reported in this study, future trials in this discipline of research ought to ensure datasets reporting of patient outcomes including follow-up. Furthermore, in an effort to improve outcomes for patients with DIPG, confounding variables including multi-disciplinary therapies must be assessed in a manner that may quantify various approaches both singularly or combined for survival (OS, EFS, PFS) and associated outcomes.”

Comment 8: There is a repetition at the end of the discussion: "There are certain limitations. There are certain limitations of this study." 

Response to Comment 8: Thank you for noticing; the repetition has been removed. Many thanks.

Reviewer 3 Report

This systematic review "Emerging Therapeutic Strategies for Diffuse Intrinsic Pontine Glioma: A Systematic Review ” summarizes a clinical picture of the last five years of the direction toward which DIPG 32 research is heading which is interesting and beneficial to the field. One suggestion is that Table 1 is too big and occupies a lot of space. Could the authors shrink it a little big?

Author Response

Before you read my responses to your comments, I humbly thank you very much for your time and due diligence in reviewing the paper. The voluntary nature of work must be acknowledged.

Regards,

Zouina S.

Comment: This systematic review "Emerging Therapeutic Strategies for Diffuse Intrinsic Pontine Glioma: A Systematic Review ” summarizes a clinical picture of the last five years of the direction toward which DIPG 32 research is heading which is interesting and beneficial to the field. One suggestion is that Table 1 is too big and occupies a lot of space. Could the authors shrink it a little big?

Author Response: Thank you for your comments and good feedback. I have reduced the length of all tables. I have also added abbreviations for reduction.